# Evaluation of outpatient treatment for non-hospitalised patients with COVID-19: The experience of a regional centre in the UK

**Amanda T. Goodwin**[1]*, **Jonathan S. Thompson**[2], **Ian P. Hall**[1]

1 NIHR Nottingham Biomedical Research Centre & University of Nottingham, Nottingham, United Kingdom,
2 Department of Infectious Diseases, Nottingham University Hospitals, Nottingham, United Kingdom

* Amanda.Goodwin@nottingham.ac.uk

**Data Availability Statement:** All relevant data are within the paper and its Supporting Information files.

## Abstract

### Introduction

Antivirals, such as molnupiravir, and SARS-CoV-2 neutralising monoclonal antibodies (nMAbs), such as sotrovimab, reduced the risk of hospitalisation and death in clinical trials of high-risk non-hospitalised patients with Covid-19. However, the real-world benefits of these drugs are unclear.

### Aims

To evaluate the characteristics and outcomes of high-risk patients referred for outpatient antiviral or nMAb treatment for symptomatic Covid-19.

### Methods

The records of patients referred to a large UK Covid Medicines Delivery Unit (CMDU) over nine weeks (December 2021-February 2022) were reviewed. Data were collected on demographics, referral indications, vaccination, deprivation, treatment, complications, hospital admission, and mortality.

### Results

1820 patients were referred to the CMDU, with 604 (33.2%) suitable for further assessment. 169 patients received sotrovimab, 80 patients received molnupiravir, 70 patients declined treatment, and 266 were ineligible for treatment because of resolving symptoms. There were trends towards higher proportions of female and white patients, lower deprivation scores, and malignancy- or transplant-related indications in the groups receiving treatment compared with untreated patients. Covid-19-related hospitalisations occurred in 1.2% of the treated group and 3.0% of the untreated group indicating a potential treatment effect, however Covid-related hospitalisations were lower than reported in the original clinical trials (2.2% compared with 7–10%).

### Conclusion

The referral pathways for outpatient treatment of Covid-19 are inefficient, and the UK system may not be serving all groups equitably. Hospitalisation with Covid-19 was rare

**Funding:** This study was funded by the National Institute for Health Research, I.P. Hall holds a NIHR Senior Investigator award (NF-SI-0617-10096); and National Institute for Health Research, ATG holds an NIHR-funded Academic Clinical Lecturer post (CL-2020-12-003).

**Competing interests:** ATG and JST - None to declare IPH has a funded research collaboration with Boehringer Ingelheim (unrelated to the topic of this paper).

regardless of treatment. Ongoing service evaluation is required to ensure efficient use of resources for the outpatient management of Covid-19.

## Introduction

Coronavirus disease 2019 (Covid-19) has been responsible to date for over 6.6 million deaths worldwide [1]. While vaccination is central to reducing the risk of hospitalisation and death due to Covid-19, there remains a significant proportion who are thought to be unable to mount a sufficient immune response to vaccination due to underlying health conditions [2]. A range of therapies have therefore been developed or repurposed to reduce the chance of progression of Covid-19 in high-risk individuals.

The two primary approaches used to reduce the risk of progression of early Covid-19 infection are treatment with neutralising monoclonal antibodies (nMAbs) and/or antiviral drugs. NMAbs and antivirals inhibit viral replication, and nMAbs also prevent SARS-Cov-2 entry into host cells, making these approaches attractive for use in early stages of Covid-19 infection when viral replication is the key driver of disease progression [2]. In the UK, non-hospitalised patients are eligible for nMAb or antiviral treatment if they have a confirmed SARS-CoV-2 infection, they are symptomatic and show no signs of clinical recovery, and are a member of the 'highest' risk group (Table 1) [3]. As of the UK National Health Service (NHS) interim clinical commissioning policy February 2022, first line treatment is nirmatrelvir/ritonavir (Paxlovid, antiviral) or sotrovimab (nMAb), however significant drug interactions and contraindications in liver and renal disease limit the use of Paxlovid [3]. Second and third line treatment are remdesivir and molnupiravir (antivirals), respectively [3].

A single dose of 500mg sotrovimab IV has been reported to give a relative risk reduction of 79% of hospitalisation or death in high-risk patients with mild-to-moderate Covid-19 [4, 5]. Molnupiravir is an oral small molecule antiviral prodrug that is active against SARS-CoV-2 [6] and has been shown to give a 30% relative risk reduction of hospitalisation or death in non-hospitalised patients with mild-moderate Covid-19 and a risk factor for severe disease when commenced within 5 days of symptom onset [6]. As these trials were performed at a time when different variants of Covid-19 were circulating to those seen now, and in very specific patient groups, real-world evaluation of treatment for early Covid-19 is urgently needed.

Antiviral and nMAb therapy must be started within 5 days of symptom onset (except for remdesivir which can be commenced within 7 days) [3], therefore efficient identification of treatment candidates and delivery of indicated medication is required. This has led to the development of NHS Covid Medicine Delivery Units (CMDUs) for patient screening and the coordination of treatment. With the frequent emergence of new variants, and the inevitable variations between real-world patients and study populations, ongoing evaluation of CMDU activities and outcomes is essential to inform policy and practice. Here we present the experience of a major CMDU serving a large population in the East Midlands of the UK.

## Methods

### Study design

This was a retrospective single centre service evaluation study of a CMDU covering mid- and South Nottinghamshire, UK. The records of all patients referred to the Nottingham University Hospitals (NUH) COVID Medicines Delivery Unit (CMDU) between 22nd December 2021 and 20th February 2022 were reviewed. Data was collected on demographics (age, sex,

**Table 1. Criteria for being "high risk" of adverse outcomes from Covid-19 [3].**

| Indication | Description |
|---|---|
| Down's syndrome | All |
| Solid cancers | Any active solid or metastatic cancer |
| | Any chemotherapy within the last 3 months |
| | Group B or C chemotherapy in the preceding 3–12 months |
| | Radiotherapy within the last 6 months |
| Haematological disease and stem cell transplant recipients | Allogeneic haematopoietic stem cell transplant (HSCT) in the last 12 months, or active graft vs host disease (GVHD) regardless of time from transplant |
| | Autologous HSCT recipient in the last 12 months |
| | Haematological malignancies who have received chimeric antigen receptor (CAR)-T cell the last 24 months, or radiotherapy in the last 6 months. |
| | Haematological malignancies receiving systemic anti-cancer treatment in the last 6 months (other than chronic myeloid leukaemia in molecular response); or first or second line tyrosine kinase inhibitors. |
| | All patients with myeloma or chronic B cell lymphoproliferative disorders |
| | Sickle cell disease |
| | Non-malignant haematological disorders receiving B cell depleting treatment within last 12 months |
| Renal disease | Renal transplants (including failed transplants within the last 12 months) |
| | Non-transplant patients who have received a comparable level of immunosuppression |
| | Chronic kidney disease (CKD) stage 4 or 5 without immunosuppression |
| Liver disease | Cirrhosis (Child's-Pugh class A, B and C) |
| | Liver transplant recipients |
| | Liver patients on immune suppressive therapy |
| Immune-mediated inflammatory disorders (IMID) | IMID with active/ unstable disease on corticosteroids, cyclophosphamide, tacrolimus, cyclosporin or mycophenolateIMID with stable disease on corticosteroids, cyclophosphamide, tacrolimus, cyclosporin, or mycophenolate |
| | IMID with active/ unstable disease including those on biologicals +/- thiopurine or methotrexate |
| | IMID treated with rituximab or other B cell depleting therapy in the last 12 months |
| Immune deficiencies | Common variable immunodeficiency |
| | Undefined primary antibody deficiency on immunoglobulin (or eligible for immunoglobulin) |
| | Hyper IgM syndromes |
| | Good's syndrome |
| | Severe combined immunodeficiency |
| | Autoimmune polyglandular syndromes/ autoimmune polyendocrinopathy, candidiasis, ectodystrophy (APECED) syndrome |
| | Primary immunodeficiency associated with impaired type I interferon signalling |
| | X-linked agammaglobulinaemia (and other primary agammaglobulinaemias) |
| | Any secondary immunodeficiency receiving, or eligible for, immunoglobulin replacement |
| HIV/AIDS | High levels of immune suppression, uncontrolled/ untreated HIV (high viral loads) or presenting acutely with AIDS defining illness |
| | On treatment for HIV with CD4 < 350 cells/mm$^3$, or CD4 > 350 cells/mm$^3$ and additional risk factor (age, diabetes, cardiovascular, obesity, liver or renal disease, homeless, alcohol dependence) |

**Table 1.** (Continued)

| Indication | Description |
|---|---|
| Solid organ transplant recipients | All |
| Rare neurological conditions | Multiple sclerosis |
| | Motor neurone disease |
| | Myasthenia gravis |
| | Huntington's disease |

IMID = immune mediated inflammatory disorders. From the UK NHS Interim clinical commissioning policy criteria for being "high risk" of adverse outcomes from Covid-19 [3]

ethnicity), indications for referral, treatment received (if any), complications of treatment, hospital admission within 28 days, and mortality. Serum anti-Spike protein antibodies were also analysed in patients receiving sotrovimab. Elective admissions for planned procedures were not counted in the admissions data. Postcodes were also collected for the calculation of deprivation scores.

Patients were referred to CMDU via several routes. This included referrals from general practitioners, the NHS 111 service, clinical specialty teams, other CMDUs, and Webview (NHS Digital). Webview is a computerised system that automatically referred patients deemed to be of high risk of severe Covid-19 to the CMDU when a positive Covid-19 PCR test was reported. It should be noted that patients could be on the list of patients deemed to be at high risk of severe Covid-19 and thus be referred to CMDU, but not have conditions that met the commissioning criteria for Covid-19 treatments. Patients were then assessed for eligibility over the telephone by a senior clinician, and arrangements made for antiviral or nMAb treatment.

Patients were eligible for treatment if they had symptomatic Covid-19 infection confirmed by PCR, were within 5 days of symptoms onset with no evidence of symptoms resolution, and had one of the conditions listed in Table 1. Patients were excluded if they required hospital admission, were already an inpatient, were outside of the 5 day treatment window, or were completely asymptomatic. Pregnant patients and patients under 16 were referred to obstetrics and paediatrics, respectively and were not included in this study.

## Ethics statement

We completed the Health Research Authority checklist (http://www.hra-decisiontools.org.uk/research), which confirmed that an ethical review board review was not required as this project is categorised as service evaluation. We also sought advice from the research governance lead at Nottingham University Hospitals Research and Innovation Department regarding the need for formal ethical approval and informed consent for this study. As this study involved the analysis of data collected routinely as part of patient care, and no patient-identifiable information was recorded, it was deemed that this study did not require formal ethical approval or informed consent processes.

## Treatment

Sotrovimab was administered as a single 500mg intravenous infusion at the CMDU, usually within 24 hours of initial screening. Patients were observed for one hour after infusion, and were followed up by telephone the following day to discuss any side effects or ongoing symptoms.

Molnupiravir was administered orally as 800mg twice a day for 5 days at home.

Nirmatrelvir/ritonavir (Paxlovid) was administered orally as 450mg/100mg twice a day for 5 days at home. As Paxlovid was launched on 10th February [7], it was not available for our entire data collection period. Patients receiving Paxlovid were therefore not included in the analysis of this study.

### Deprivation scores

For deprivation data, lower layer super output areas were identified from post codes using the UK Government 2019 Indices of Deprivation database [8]. The index of multiple deprivation (IMD) from each area was obtained from an open access resource provided by the Ministry of Housing [9].

### Statistical analyses

Comparisons were made between the treatment groups (received sotrovimab, received molnupiravir, resolving symptoms, and declined treatment) and between those receiving treatment (sotrovimab and molnupiravir) and those not receiving treatment but within the 5 day treatment window (resolving symptoms and declined treatment). For statistical analyses, GraphPad Prism 8.1.2 was used. Data were tested for normality and descriptive statistics obtained. For comparisons between two groups, a Mann-Whitney test was performed. For comparisons between three or more groups, a Kruskall-Wallis test was performed. For the calculation of the odds ratio, the Bapista-Pike method was used.

## Results

### Treatment decisions

During the study period, 1820 patients were screened by the CMDU, of whom 604 progressed to more detailed treatment discussions (33.2%) (Fig 1). Of these potentially eligible patients, 254 received treatment (42.1%), 70 declined treatment despite being eligible (11.5%), and 266 (44.0%) reported resolving symptoms and therefore were ineligible for treatment. It was unclear from the records whether 14 (2.3%) patients received any treatment, therefore these patients were excluded from further analyses.

In total, 249 patients received either sotrovimab or molnupiravir (76.9% of those eligible for treatment after full screening). Sotrovimab (169, 66.5% of those receiving treatment, and molnupiravir (80, 31.5%) were the main treatments prescribed. Another five patients received Paxlovid (2% of those receiving treatment), but were not included in further analyses because Paxlovid was not available for the entire study period. Details of the patients that received Paxlovid can be found in the supplementary information (S1 Table). During early 2022, evidence emerged of superior efficacy of sotrovimab compared with molnupiravir, and sotrovimab became the preferred treatment approach unless IV infusion or attendance at CMDU was not possible [3].

### Patient demographics and indications for treatment

The demographic data and indications for treatment are shown in Table 2. The median age was comparable between the treated and untreated groups (51–52 years), with a trend towards a higher proportion of female patients in the treated group (62.1% of sotrovimab-treated and 69.4% of molnupiravir-treated) compared with the untreated group (57.1% of those that declined treatment, and 58.6% of those with resolving symptoms). Ethnicity data were not available for all patients, in particular the group with resolving symptoms, therefore a full

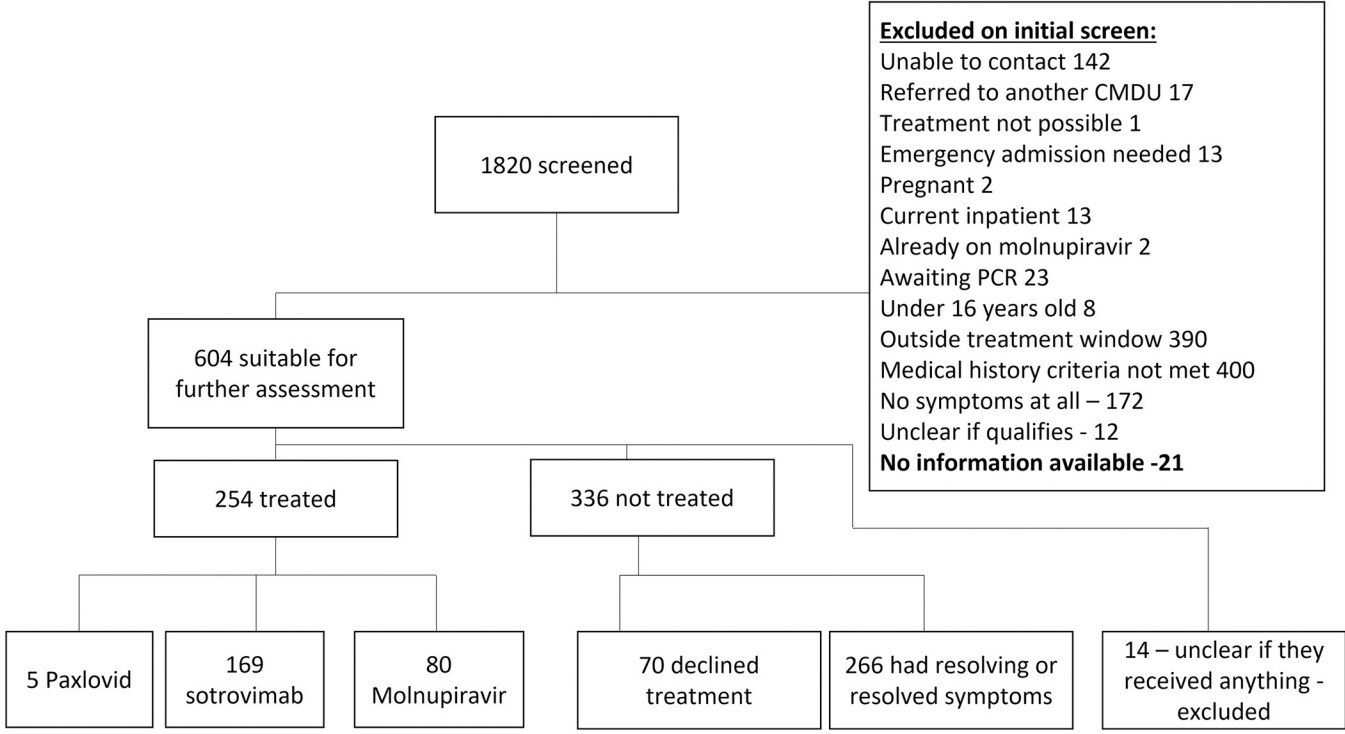

**Fig 1. The Covid Medicines Delivery Unit referral pathway.** Flowchart showing the screening results of the 1820 patients referred to the Covid Medicines Delivery Unit (CMDU).

analysis of the differences between ethnic groups could not be performed. However, there was an apparent trend towards more patients from white backgrounds receiving treatment than those from non-white ethnic groups.

The distribution of criteria making our cohort of patients "high risk" is shown in Table 2. Immune-mediated inflammatory disorders (IMIDs) were the predominant indication for referral for treatment. There was a non-significant trend towards more patients having malignancy- (either haematological or solid organ) or transplant-related indications for treatment in the treated group.

Vaccination status was recorded for 180 patients across the "treated" and "declined treatment" groups. Of these, 172 (95.6%) had received at least two Covid-19 vaccination doses. It was not possible to further analyse the relationship between vaccination status, treatment decisions, and patient characteristics and outcomes because of a large amount of missing data and the small numbers in our study.

## Sotrovimab is safe and well-tolerated

Patients receiving sotrovimab were followed up 24 hours after the infusion, and side effects recorded. Follow up was recorded for 165/169 patients (97.6%), three (1.7%) could not be contacted, and one (0.6%) had no attempt at follow up recorded. Of the patients with whom contact could be made, 12 reported possible side effects (7.3%), lower than the 10% reported in the product literature. Five patients complained of nausea (3.0%), three reported dizziness (1.8%), two reported a rash (2.4%), and there was one report each of mild confusion, palpitations, itch, shivering, and bradycardia with low blood pressure (0.6%). No side effects were reported as severe, and none required hospital admission. Sotrovimab was therefore well-tolerated in our patient population.

**Table 2. Demographic details, indications for treatment, and vaccination status of patients screened within 5 days of symptoms onset.**

| | Sotrovimab | Molnupiravir | Declined treatment | Resolving symptoms |
|---|---|---|---|---|
| N | 169 | 80 | 70 | 266 |
| **Median age (range)** | 52 (17–91) | 51 (20–85) | 51 (16–89) | 52 (16–90) |
| **Female (%)** | 105 (62.1%) | 50 (69.4%) | 41 (58.6%) | 152 (57.1%) |
| **Ethnicity** | | | | |
| White British/ Irish/ Other white background | 121 (71.6%) | 51 (63.8%) | 48 (68.6%) | 159 (59.8%) |
| Black background | 3 (1.8%) | 3 (3.8%) | 3 (4.3%) | 13 (4.9%) |
| Asian background | 8 (4.7%) | 1 (1.3%) | 4 (5/7%) | 5 (1.9%) |
| Mixed background | 5 (3.0%) | 1 (1.3%) | 2 (2.9%) | 5 (1.9%) |
| Other | 2 (1.2%) | 0 (0%) | 0 (0%) | 5 (1.9%) |
| Unknown | 30 (17.8%) | 24 (30%) | 13 (18.6%) | 79 (29.7%) |
| **Main medical indication for treatment** | | | | |
| Haematological malignancy | 27 (16.0%) | 9 (11.3%) | 6 (8.6%) | 16 (6.0%) |
| Solid organ or metastatic malignancy | 9 (5.3%) | 0 (0%) | 1 (1.4%) | 6 (2.3%) |
| Chemotherapy or radiotherapy meeting policy criteria | 6 (3.6%) | 5 (6.3%) | 6 (8.6%) | 9 (3.4%) |
| Non-transplant renal disease | 10 (5.9%) | 5 (6.3%) | 1 (1.4%) | 10 (3.8%) |
| Chronic liver disease | 8 (4.7%) | 3 (3.8%) | 2 (2.9%) | 2 (0.8%) |
| Unknown indication | 7 (4.1%) | 4 (5%) | 14 (20%) | 155 (58.3%) |
| HIV | 1 (0.6%) | 1 (1.3%) | 1 (1.4%) | 3 (1.1%) |
| Chronic neurological condition | 18 (10.7%) | 17 (21.3%) | 8 (11.4%) | 24 (9.0%) |
| Immune mediated inflammatory condition (IMID) | 67 (39.6%) | 27 (33.8%) | 24 (34.3%) | 33 (12.4%) |
| Solid organ transplant | 14 (8.3%) | 6 (7.5%) | 4 (5.7%) | 3 (1.1%) |
| Primary immunodeficiency | 2 (1.2%) | 3 (3.8%) | 3 (4.3%) | 5 (1.9%) |
| **Vaccination status–actual number of doses received** | | | | |
| 0 | 2 (1.2%) | 1 (1.3%) | 3 (4.3%) | - |
| 1 | 0 (0%) | 1 (1.3%) | 1 (1.4%) | - |
| 2 | 6 (3.6%) | 4 (5%) | 2 (2.9%) | - |
| 3 | 96 (56.8%) | 17 (21.3%) | 28 (40%) | - |
| 4 | 19 (11.2%) | 0 (0%) | 0 (0%) | - |
| Unknown | 46 (27.2%) | 57 (71.3%) | 36 (51.4%) | - |
| **Vaccination status–grouped by 2 doses or less, and 3 doses or more** | | | | |
| 2 doses or less | 8 (4.7%) | 6 (7.5%) | 6 (8.6%) | - |
| 3 doses or more | 115 (68%) | 17 (21.3%) | 28 (40%) | - |

IMID = immune mediated inflammatory disorder. Data on vaccination status not available for those with resolving symptoms.

## Patient outcomes

Hospital admissions and mortality data are shown in Table 3. 31 (5.2%) of the 599 patients screened within five days of symptom onset had an unplanned hospital admission within 28 days, of which 13 (2.2%) were Covid-19 related. Two patients from the sotrovimab group and one from the molnupiravir group (1.2% and 1.3% of each group, respectively) had a Covid-19-related hospital admission, compared with three of those that declined treatment (4.3%) and seven that reported resolving symptoms at screening (2.6%). This gives an overall 3.0% rate of admission within 28 days due to Covid 19 in the untreated group, compared with 1.2% in the treated group. These data suggest a trend towards a reduction in hospitalisation with treatment for early Covid-19 in high-risk patients but because rates of Covid-related hospital admission were low in all groups no significant effect of treatment was seen (odds ratio of Covid-19-related hospital admission of 0.40, 95% CI 0.12–1.36).

**Table 3. 28 day hospital admission and mortality.**

|  | Sotrovimab | Molnupiravir | Declined treatment | Resolving symptoms |
|---|---|---|---|---|
| N | 169 | 80 | 70 | 266 |
| Hospital admission within 28 days | 11 (6.5%) | 2 (2.5%) | 3 (4.3%) | 15 (5.6%) |
| Covid-related hospital attendance within 28 days | 2 (1.2%) | 1 (1.3%) | 3 (4.3%) | 7 (2.6%) |
| Mortality within 28 days | 0 (0%) | 1 (1.3%) | 0 (0%) | 2 (0.8%) |

Three patients who were screened within 5 days of Covid-19 symptom onset died within 28 days of screening (0.5%). All three deaths were related to Covid-19. One death occurred in the molnupiravir group, and two deaths occurred in the "resolving symptoms" group. There were no deaths within 28 days of screening in either the sotrovimab or "declined treatment" groups.

The presence of endogenous anti-SARS-CoV antibodies was analysed in patients receiving sotrovimab prior to their infusions. Results were available for 152 patients that received sotrovimab (90%). Of these, 21 (13.8%) had undetectable antibody levels, and 131 (86.2%) had detectable antibodies. In a subgroup of 68 samples where antibodies were quantified 32 (47.1%) only had low antibody levels. Of the 169 patients treated with sotrovimab, 123 had their vaccination status recorded and 121 (98.4%) of these had received at least two Covid-19 vaccine doses (Table 2), and these low antibody levels could represent a suboptimal antibody response to vaccination. However, given the small numbers of hospital admissions and mortality in this group, it was not possible to assess the effect of antibody status or vaccination on outcomes. However, these data indicate that many patients deemed at high risk of poor outcomes from Covid-19 can generate antibodies to either vaccination or natural infection, although in at least some these are at a lower level than in the general population.

## Patient deprivation may influence treatment decisions

Index of multiple deprivation (IMD) data were generated from the home postcodes of patients who progressed past the initial screening stage (Table 4). Postcodes were available for all patients except one in the "resolving symptoms group". The median IMD score was lower in the treated groups, at 16.1 (range (1.1–73.3) and 16.2 (1.6–70.4) in the sotrovimab and molnupiravir groups, respectively, compared with 20.9 (range 1.3–70.4) and 21.9 (range 1.1–72.2) in those that declined treatment or had resolving symptoms. When patients that received any treatment (sotrovimab or molnupiravir) were compared to those that did not receive treatment (declined treatment or reported resolving symptoms), the deprivation score was significantly higher in those that were not treated (IMID 21.8 in the untreated group compared with 16.1 in the treated group, p = 0.02, Mann-Whitney test). These data imply that patients experiencing higher degrees of deprivation are less likely to access treatment for early Covid-19.

**Table 4. Index of multiple deprivation data of patients referred to the Covid medicines delivery unit.**

|  | Sotrovimab | Molnupiravir | Declined treatment | Resolving symptoms |
|---|---|---|---|---|
| N | 169 | 80 | 70 | 266 |
| Mean IMD | 20.61 | 22.00 | 21.89 | 24.42 |
| Median IMD | 16.12 | 16.16 | 20.90 | 21.94 |
| IMD range | 1.1–73.3 | 1.6–70.4 | 1.3–70.4 | 1.1–72.2 |

IMD = index of multiple deprivation

## Discussion

The data presented here provide a valuable early insight into patient outcomes seen in a UK CMDU over a 10 week period in early 2022. Importantly, only 13.7% of patients initially screened received either molnupiravir or sotrovimab treatment. This implies that the current system for identifying potentially eligible patients is inefficient, particularly as 400 patients (22%) were found not to have a medical indication for treatment. This may be because the automated referral systems flagged patients to the CMDU that were deemed high risk of severe Covid-19, but may not have met the commissioning criteria for treatment at the time of this study [3], for example patients with diabetes. Furthermore, 390 patients (21.4%) were already outside of the treatment window by the time contact was made with CMDU. We are aware that, particularly in the early stages of the CMDU service, many patients were screened by CMDU beyond the first five days of symptoms, and were thus ineligible for treatment. This issue was multifactorial, and likely related to the introduction of new clinical and computer systems, lower awareness of treatments among the patients and healthcare professionals, and the timeliness of PCR results. Further work is required to publicise the eligibility criteria so that clinical teams and patients are aware of the indications for treatment, and to facilitate timely referral.

In our patient cohort, we had low rates of Covid-19-related hospitalisation (2.2% of all patients within 28 days of CMDU triage). This is lower than the 7% 28 day hospitalisation or mortality rate reported in the placebo arm of the sotrovimab trials [4, 5], and 10% reported in the molnupiravir trial [6]. Similarly, only three deaths occurred in the 604 eligible patients presenting within the treatment window (0.5% 28 day mortality rate across all groups). This may reflect the impact of vaccination even in these at risk groups and less severe disease seen with variants circulating in early 2022 in the UK (predominantly Omicron and BA1) [10]. These real-world data however raise the question of the cost-effectiveness of the treatments given in CMDUs. As of the week ending 1st May 2022, 41649 patients have received either antiviral or nMAb treatment for Covid-19 [7], therefore is it imperative to understand how these treatments affect patient outcomes.

We observed a trend towards reduced hospitalisation rates in the treated patients compared to patients that did not receive treatment. This did not reach statistical significance, which is likely due to our study being underpowered for detecting such differences between treatment groups. Assuming equal matching between the treatment groups, power to detect a difference between groups of 80%, and an $\alpha$ value (probably of type I error) of 0.05, the required number of subjects to detect a significant difference in outcome based on the event rates we observed would be 1990. Therefore, it is not possible to conclude whether treatment for early Covid-19 reduced the rates of hospitalisation in our study population, which was significantly smaller than this value.

In our study, 266 patients were excluded from treatment because they reported resolving symptoms. Whether symptoms are "resolving", "persistent", or "worsening" is subjective from both the patient and healthcare professional perspectives. Differences in symptom reporting and healthcare seeking habits of different patient groups could result in significant differences in access to treatments for Covid-19. While there were no statistically significant differences in outcomes between treated and untreated groups in our study, the two deaths and seven Covid-related hospital admissions in those with "resolving symptoms" suggests that this may not be an appropriate indication for exclusion from treatment for Covid-19. Alternative non-subjective scoring systems for the persistence or resolution of Covid-19 symptoms are required to address this issue.

A further explanation for the lower hospitalisation and mortality rates we observed may be because of differences between the study populations. In the original sotrovimab and

molnupiravir trials [4–6], the patient population was deemed "high-risk" because of age (over 55 years), obesity, diabetes, heart failure, or chronic lung disease [4, 5]. Our study population was deemed "high-risk" according to the interim clinical commissioning policy (Table 1) [3], which consist of different criteria to those used in the clinical trials. These differences in baseline mortality may therefore reflect a difference in patients' underlying risks. Alternatively, these differences could be driven by different uptake of vaccination, as being unvaccinated was a requirement for inclusion in the molnupiravir trial [6]. We could not fully analyse vaccination status in our population, as there was a significant amount of missing data, however many patients had received at least one vaccination. Studies of other CMDUs in the NHS, and similar units across the world, will help to clarify the baseline risk of hospitalisation and death in these "high-risk" patients.

The commonest complications of sotrovimab were nausea and dizziness, in keeping with the product literature [3], and no severe reactions occurred. These data support the finding of clinical trials and real-world studies that sotrovimab is a safe treatment for early Covid-19 [4, 5, 11, 12].

We also assessed baseline demographic differences between those accepting treatment or not receiving treatment. Although our study was small there was a trend towards a higher proportion of patients receiving treatment being female and from white ethnic backgrounds. This may reflect differences in treatment-seeking behaviours, underlying disease, or symptom manifestation between the sexes and different ethnic groups. The deprivation data suggest that patients receiving treatment had lower levels of deprivation than those that were not treated. Unfortunately, the reasons for declining treatment were infrequently recorded. Possible explanations include patients from more deprived background being less likely to be able to attend for a sotrovimab infusion, being more reluctant to travel for treatment, or being more likely to minimise their symptoms and decline treatment. Furthermore, patients from more deprived backgrounds may have had different prior experiences of healthcare, and may be less trusting of and therefore less willing to accept the relatively new treatments offered for Covid-19. This is a concerning finding given the universal healthcare model in the UK's NHS, but is consistent with other reports that patients from more deprived areas may receive poorer quality healthcare [13]. Studies in the NHS and other healthcare systems may clarify the reasons behind this. Further consideration is required to ensure that hard to reach groups are not excluded from treatment for early Covid-19.

Our study is limited by being a retrospective review of the medical records of a local population, and missing data has limited our ability to address issues such as antibody status. Expansion of this work to include other areas in the UK would strengthen our conclusions, and routine prospective gathering of the data analysed here will facilitate future service evaluations.

Omicron became the dominant variant in England mid December 2022 and is likely to have been the predominant Covid-19 variant during our study [10]. While sotrovimab retained efficacy against the original omicron strain, the BA.2 subvariant exhibited increased resistance to sotrovimab in vitro [14]. Although the now predominant BA.4 and BA.5 strains are not thought to have further reduced susceptibility to sotrovimab [15], this drug is no longer recommended for use by the US Food and Drug Administration (FDA) [16]. This highlights the need for ongoing monitoring of nMAbs and antiviral treatment for this rapidly evolving virus, and the continued development and assessment of new treatment strategies.

## Conclusion

Current referral pathways for outpatient treatment of early Covid-19 in the UK are inefficient, and ongoing service evaluation is required to ensure that all eligible patients are offered

treatment if the service continues. Hospitalisation with Covid-19 was rare in our population regardless of whether treatment was received. The population highlighted as high risk by NHS clinical commissioning has different underlying disease profiles and vaccination uptake rates to those assessed in the sotrovimab and molnupiravir clinical trials. More work is required to ensure hard to reach groups, which may include ethnic minority populations and deprived individuals, have equitable access to treatment of early Covid-19.

## Supporting information

**S1 Table. Demographic and clinical details of patients that received Paxlovid.**
(DOCX)

## Acknowledgments

We would like to thank Ms Judith Palmer (Chief Pharmacist, Nottingham University Hospitals) for advice on obtaining the CMDU data, and Dr Jeremy Lewis (Consultant Physician, Nottingham University Hospitals) for assistance in identifying patients for this work.

## Author Contributions

**Conceptualization:** Amanda T. Goodwin, Jonathan S. Thompson, Ian P. Hall.

**Data curation:** Amanda T. Goodwin, Jonathan S. Thompson.

**Formal analysis:** Amanda T. Goodwin.

**Investigation:** Amanda T. Goodwin, Jonathan S. Thompson, Ian P. Hall.

**Methodology:** Amanda T. Goodwin, Jonathan S. Thompson, Ian P. Hall.

**Writing – original draft:** Amanda T. Goodwin, Ian P. Hall.

**Writing – review & editing:** Amanda T. Goodwin, Jonathan S. Thompson, Ian P. Hall.

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
