## [Decision Letter · Decision Letter 0]

8 Nov 2022

PONE-D-22-19820Evaluation of outpatient treatment for non-hospitalised patients with COVID-19 in the UKPLOS ONE

Dear Dr. Amanda T Goodwin,

Thank you for submitting your manuscript to PLOS ONE. After careful consideration, we feel that it has merit but does not fully meet PLOS ONE’s publication criteria as it currently stands. Therefore, we invite you to submit a revised version of the manuscript that addresses the points raised during the review process.

We look forward to receiving your revised manuscript.

Kind regards,

Seung-Hwa Lee

Academic Editor

PLOS ONE

2. You indicated that ethical approval was not necessary for your study. We understand that the framework for ethical oversight requirements for studies of this type may differ depending on the setting and we would appreciate some further clarification regarding your research. Could you please provide confirmation from your institutional review board or research ethics committee (e.g., in the form of a letter or email correspondence) that ethics review was not necessary for this study? Please include a copy of the correspondence as an "Other" file.

4. Thank you for stating the following in the Funding Section of your manuscript:

“ATG holds an NIHR-funded Academic Clinical Lecturer post. IPH holds an NIHR Senior Investigator Award.”

“ATG holds an NIHR-funded Academic Clinical Lecturer post. IPH holds an NIHR Senior Investigator Award. The funders had no role in study design, data collection and analysis, decision to publish, or preparation of the manuscript.”

“ATG and JST - None to declare

IPH has a funded research collaboration with Boehringer Ingelheim (unrelated to the topic of this paper)”

Reviewers' comments:

Reviewer's Responses to Questions

**Comments to the Author**

1. Is the manuscript technically sound, and do the data support the conclusions?

Reviewer #1: Yes

Reviewer #2: Yes

2. Has the statistical analysis been performed appropriately and rigorously? 

Reviewer #1: Yes

Reviewer #2: Yes

3. Have the authors made all data underlying the findings in their manuscript fully available?

Reviewer #1: Yes

Reviewer #2: Yes

4. Is the manuscript presented in an intelligible fashion and written in standard English?

Reviewer #1: Yes

Reviewer #2: Yes

5. Review Comments to the Author

Reviewer #1: The current issue of the benefits from antivirals and nMAbs against SARS CoV-2 is extensively treated in this manuscripts, no final conclusions can be drawn due to the many limitations and caveats of the study correctly considered by the authors.

What is striking is that, despite the large amount of patients potentially eligible, only for 1 third of them an assessment for treatment was done.

This very low rate of suitable patients for assessment uncovers criticisms on referral pathway. This point has been underlined by the authors but some points need to be clarified: as for M&M it could be useful to clarify in the last paragraph of "Study design"if the process is computerized, if being within a high risk group is an information currently available for all the patients or physicians have to retrive this info case-by-case and which is the timing for registration/transfer of info to CMDU and so on.

Finally, due to the poor external validity of these results I suggeste to modify the title in this way:

"Evaluation of outpatient treatment for non-hospitalised patients with COVID-19: a mono-institutional experience in UK"

Reviewer #2: This is a retrospective review of UK experience of treating high risk immunosuppressed individuals with sotrovimab or molnupiravir for COVID-19. The experience is from December 2021 to Feb 2022 and so looks at the impact of Omicron BA1. At this time the majority were dual or better vaccinated. The criteria are based on UK guidelines for these agents and are already quite restrictive to people with high order immunosuppression.

This is important data, as most governments have embraced the use of these agents in their vaccinated populations despite no evidence of efficacy or cost effectiveness.

Both mortality and hospitalisation were rare and the analysis is probably under powered to detect a difference.

Impact is also lessened by the change in variants and the fact that Sotrovimab is likely no longer effect with current variants, in addition Paxlovid with probably better efficacy would be a more valuable intervention to assess, but this is out of the control of the authors and the data still has value.

Major points:

1. The 2 most important points, the first could be highlighted more is that resolving symptoms is an exclusion criteria. 266 were excluded with resolving symptoms, given the subjective nature of this, 2 deaths in the resolving symptoms group, and 15 admissions at d28 for COVID suggests this should not be an exclusion criteria for treatment. This could be made clearer.

2. The fact that those from more deprived backgrounds are also less likely to receive treatment is important but well highlighted. Can the authors give any further insight why?

3. Admission at d28 was 3% in untreated vs 1.2% in treated

with an OR 0.4 (95%CI 0.12 to 1.36). Can the authors calculate/estimate the sample size they would need to see an effect with this event rate?

4. Table 2 in vaccination status, would be helpful to add categories for more than x2 and more than x3 vaccinations with percentages.

6. PLOS authors have the option to publish the peer review history of their article (what does this mean?). If published, this will include your full peer review and any attached files.

Reviewer #1: No

Reviewer #2: **Yes: **Peter AB Wark

---

## [Author Response · Author response to Decision Letter 0]

15 Dec 2022

Many thanks for your feedback on our manuscript. Please find attached a revised manuscript addressing each of the reviewer and editor comments, and details of how we have responded to these comments below. 

Journal Requirements:

1) We have reformatted our manuscript to meet PLOS ONE’s style requirements 

2) We have uploaded correspondence from the research governance team in the Research and Innovation department of Nottingham University Hospitals confirming that ethical review was not required for this work.

3) The “funding information” section on the submission portal has been updated with award numbers. Please could the funding statement be recorded as below:

“ATG holds an NIHR-funded Academic Clinical Lecturer post. IPH holds an NIHR Senior Investigator Award (award number NF-SI-0617-10096). The funders had no role in study design, data collection and analysis, decision to publish, or preparation of the manuscript.”

Please note, the ACL post held by ATG is not associated with an award number, therefore this is not stated.

4) The “funding information” section of the manuscript has been removed

5) The “competing interests” section has been updated to confirm that our declarations of interest do not alter our adherence to PLOS ONE policies on sharing data and materials. This statement should now read as below:

“Competing interests: ATG and JST - None to declare. IPH has a funded research collaboration with Boehringer Ingelheim (unrelated to the topic of this paper). This does not alter our adherence to PLOS ONE policies on sharing data and materials.”

6) We have reviewed the reference list, which is complete and correct. Reference 1 has been edited slightly to ensure that the reported number of COVID-19 related deaths is up to date (edited from 6.2 to 6.6 million deaths worldwide in the opening paragraph of the introduction). The URLs in references 8, 10, and 15 (original numbers) have been updated and we confirm that these led to the correct information on 1st December 2022. References 11-13 (original numbers) have been updated with the full citations (i.e. page numbers); this information was unavailable at the time of the original submission. We have also added an additional reference to address Reviewer 2’s comment on the deprivation data. 

Reviewer comments

Reviewer 1

1) Reviewer comment: “ This very low rate of suitable patients for assessment uncovers criticisms on referral pathway. This point has been underlined by the authors but some points need to be clarified: as for M&M it could be useful to clarify in the last paragraph of "Study design" if the process is computerized, if being within a high risk group is an information currently available for all the patients or physicians have to retrive this info case-by-case and which is the timing for registration/transfer of info to CMDU and so on”

Response: We agree that our study reveals inefficiencies in the referral pathway to CMDU. We have added more detail about the CMDU referral pathways to the “study design” section of the revised manuscript. Additions have also been made to the first paragraph of the discussion to expand upon possible reasons for the inefficiencies that we observed. 

2) Reviewer comment: “Finally, due to the poor external validity of these results I suggest to modify the title in this way: "Evaluation of outpatient treatment for non-hospitalised patients with COVID-19: a mono-institutional experience in UK"”

Response: We agree with this comment, and have edited our title to “Evaluation of outpatient treatment for non-hospitalised patients with COVID-19: The experience of a regional centre in the UK”. We have opted to call this a “regional” study rather than mono-institutional, as our CMDU covers a large region of mid- and south Nottinghamshire in the UK, which encompasses areas covered by several different NHS hospital trusts. 

Reviewer 2

1) Reviewer comment: “Resolving symptoms is an exclusion criteria. 266 were excluded with resolving symptoms, given the subjective nature of this, 2 deaths in the resolving symptoms group, and 15 admissions at d28 for COVID suggests this should not be an exclusion criteria for treatment. This could be made clearer”

Response: Thank you for this comment. We have added a section to the discussion (3rd paragraph of discussion) to address this. We have referred to the 7 covid-related hospital attendances rather than the 15 hospital admissions within 28 days, as some of these admissions were deemed to be unrelated to Covid-19. 

2) Reviewer comment: “The fact that those from more deprived backgrounds are also less likely to receive treatment is important but well highlighted. Can the authors give any further insight why?”

Response: Unfortunately, the reasons that patients gave for declining treatment were not frequently recorded, and thus we could not analyse this in more detail. We suspect that patients from more deprived backgrounds are less likely to be able to attend for a sotrovimab infusion (for example time off work or childcare), are more reluctant to travel for treatment, or they may be more likely to minimise their symptoms and decline treatment. Furthermore, patients from more deprived backgrounds may be less trusting of and therefore less willing to accept these relatively new treatments. The reasons behind inequitable access to treatment are an important area of further research, and the Nuffield Trust has reported that patients from areas of greater deprivation are more likely to receive poorer quality care across a range of healthcare services (Scobie S, Morris J. Quality and inequality: digging deeper. Nuffield Trust and the Health Foundation. 2020. https://www.nuffieldtrust.org.uk/public/files/2020-01/quality_inequality.) We have edited paragraph 6 of the discussion to emphasise this further, and added the reference to the study by Scobie et al. 

3) Reviewer comment: “Admission at d28 was 3% in untreated vs 1.2% in treated with an OR 0.4 (95%CI 0.12 to 1.36). Can the authors calculate/estimate the sample size they would need to see an effect with this event rate?”

Response: Many thanks for this suggestion. On our power calculations, we estimate that, assuming equal matching in the groups and power set at 80%, the required number of subjects to detect a significant difference in outcome based on the event rates we observed would be 1990. We have added this to the manuscript discussion. 

4) “Table 2 in vaccination status, would be helpful to add categories for more than x2 and more than x3 vaccinations with percentages”

We have edited Table 2 to include the percentages of patients who had received 1, 2, 3, or 4 doses of a Covid-19 vaccination, and separate section for those that received two doses or less / 3 doses or more of a Covid-19 vaccination.

Many thanks for your continued consideration of our manuscript for publication.

---

## [Editor Report · Decision Letter 1]

5 Jan 2023

PONE-D-22-19820R1Evaluation of outpatient treatment for non-hospitalised patients with COVID-19: The experience of a regional centre in the UKPLOS ONE

Dear Dr. Amanda T Goodwin

Thank you for submitting your manuscript to PLOS ONE. After careful consideration, we feel that it has merit but does not fully meet PLOS ONE’s publication criteria as it currently stands. Therefore, we invite you to submit a revised version of the manuscript that addresses the points raised during the review process.

We look forward to receiving your revised manuscript.

Kind regards,

Seung-Hwa Lee

Academic Editor

PLOS ONE

Journal Requirements:

Reviewers' comments:

Here is response from Reviewer 1

The current issue of the benefits from antivirals and nMAbs against SARS CoV-2 is extensively treated in this manuscripts, no final conclusions can be drawn due to the many limitations and caveats of the study correctly considered by the authors.

What is striking is that, despite the large amount of patients potentially eligible, only for 1 third of them an assessment for treatment was done.

This very low rate of suitable patients for assessment uncovers criticisms on referral pathway. This point has been underlined by the authors but some points need to be clarified: as for M&M it could be useful to clarify in the last paragraph of "Study design"if the process is computerized, if being within a high risk group is an information currently available for all the patients or physicians have to retrive this info case-by-case and which is the timing for registration/transfer of info to CMDU and so on.

Finally, due to the poor external validity of these results I suggeste to modify the title in this way:

"Evaluation of outpatient treatment for non-hospitalised patients with COVID-19: a mono-institutional experience in UK"

and Review 2

This is a retrospective review of UK experience of treating high risk immunosuppressed individuals with sotrovimab or molnupiravir for COVID-19. The experience is from December 2021 to Feb 2022 and so looks at the impact of Omicron BA1. At this time the majority were dual or better vaccinated. The criteria are based on UK guidelines for these agents and are already quite restrictive to people with high order immunosuppression.

This is important data, as most governments have embraced the use of these agents in their vaccinated populations despite no evidence of efficacy or cost effectiveness.

Both mortality and hospitalisation were rare and the analysis is probably under powered to detect a difference.

Impact is also lessened by the change in variants and the fact that Sotrovimab is likely no longer effect with current variants, in addition Paxlovid with probably better efficacy would be a more valuable intervention to assess, but this is out of the control of the authors and the data still has value.

Major points:

1. The 2 most important points, the first could be highlighted more is that resolving symptoms is an exclusion criteria. 266 were excluded with resolving symptoms, given the subjective nature of this, 2 deaths in the resolving symptoms group, and 15 admissions at d28 for COVID suggests this should not be an exclusion criteria for treatment. This could be made clearer.

2. The fact that those from more deprived backgrounds are also less likely to receive treatment is important but well highlighted. Can the authors give any further insight why?

3. Admission at d28 was 3% in untreated vs 1.2% in treated

with an OR 0.4 (95%CI 0.12 to 1.36). Can the authors calculate/estimate the sample size they would need to see an effect with this event rate?

4. Table 2 in vaccination status, would be helpful to add categories for more than x2 and more than x3 vaccinations with percentages.

---

## [Author Response · Author response to Decision Letter 1]

10 Jan 2023

Dear Dr Lee and the editorial board of PLOS-ONE,

Many thanks for your feedback on our manuscript. Please find attached a revised manuscript addressing each of the reviewer and editor comments, and details of how we have responded to these comments below. 

Journal Requirements:

1) We have reformatted our manuscript to meet PLOS ONE’s style requirements 

2) We have uploaded correspondence from the research governance team in the Research and Innovation department of Nottingham University Hospitals confirming that ethical review was not required for this work.

3) The “funding information” section on the submission portal has been updated with award numbers. Please could the funding statement be recorded as below:

“ATG holds an NIHR-funded Academic Clinical Lecturer post (CL-2020-12-003). IPH holds an NIHR Senior Investigator Award (award number NF-SI-0617-10096). The funders had no role in study design, data collection and analysis, decision to publish, or preparation of the manuscript.”

4) The “funding information” section of the manuscript has been removed

5) The “competing interests” section has been updated to confirm that our declarations of interest do not alter our adherence to PLOS ONE policies on sharing data and materials. This statement should now read as below:

“Competing interests: ATG and JST - None to declare. IPH has a funded research collaboration with Boehringer Ingelheim (unrelated to the topic of this paper). This does not alter our adherence to PLOS ONE policies on sharing data and materials.”

6) We have reviewed the reference list, which is complete and correct, and does not contain any retracted papers. Reference 1 has been edited slightly to ensure that the reported number of COVID-19 related deaths is up to date (edited from 6.2 to 6.6 million deaths worldwide in the opening paragraph of the introduction). The URLs in references 8, 10, and 15 (original numbers) have been updated and we confirm that these led to the correct information on 1st December 2022. References 11-13 (original numbers) have been updated with the full citations (i.e. page numbers); this information was unavailable at the time of the original submission. We have also added an additional reference to address Reviewer 2’s comment on the deprivation data. 

Reviewer comments

Reviewer 1

1) Reviewer comment: “ This very low rate of suitable patients for assessment uncovers criticisms on referral pathway. This point has been underlined by the authors but some points need to be clarified: as for M&M it could be useful to clarify in the last paragraph of "Study design" if the process is computerized, if being within a high risk group is an information currently available for all the patients or physicians have to retrive this info case-by-case and which is the timing for registration/transfer of info to CMDU and so on”

Response: We agree that our study reveals inefficiencies in the referral pathway to CMDU. We have added more detail about the CMDU referral pathways to the “study design” section of the revised manuscript. Additions have also been made to the first paragraph of the discussion to expand upon possible reasons for the inefficiencies that we observed. 

2) Reviewer comment: “Finally, due to the poor external validity of these results I suggest to modify the title in this way: "Evaluation of outpatient treatment for non-hospitalised patients with COVID-19: a mono-institutional experience in UK"”

Response: We agree with this comment, and have edited our title to “Evaluation of outpatient treatment for non-hospitalised patients with COVID-19: The experience of a regional centre in the UK”. We have opted to call this a “regional” study rather than mono-institutional, as our CMDU covers a large region of mid- and south Nottinghamshire in the UK, which encompasses areas covered by several different NHS hospital trusts. 

Reviewer 2

1) Reviewer comment: “Resolving symptoms is an exclusion criteria. 266 were excluded with resolving symptoms, given the subjective nature of this, 2 deaths in the resolving symptoms group, and 15 admissions at d28 for COVID suggests this should not be an exclusion criteria for treatment. This could be made clearer”

Response: Thank you for this comment. We have added a section to the discussion (3rd paragraph of discussion) to address this. We have referred to the 7 covid-related hospital attendances rather than the 15 hospital admissions within 28 days, as some of these admissions were deemed to be unrelated to Covid-19. 

2) Reviewer comment: “The fact that those from more deprived backgrounds are also less likely to receive treatment is important but well highlighted. Can the authors give any further insight why?”

Response: Unfortunately, the reasons that patients gave for declining treatment were not frequently recorded, and thus we could not analyse this in more detail. We suspect that patients from more deprived backgrounds are less likely to be able to attend for a sotrovimab infusion (for example time off work or childcare), are more reluctant to travel for treatment, or they may be more likely to minimise their symptoms and decline treatment. Furthermore, patients from more deprived backgrounds may be less trusting of and therefore less willing to accept these relatively new treatments. The reasons behind inequitable access to treatment are an important area of further research, and the Nuffield Trust has reported that patients from areas of greater deprivation are more likely to receive poorer quality care across a range of healthcare services (Scobie S, Morris J. Quality and inequality: digging deeper. Nuffield Trust and the Health Foundation. 2020. https://www.nuffieldtrust.org.uk/public/files/2020-01/quality_inequality.) We have edited paragraph 6 of the discussion to emphasise this further, and added the reference to the study by Scobie et al. 

3) Reviewer comment: “Admission at d28 was 3% in untreated vs 1.2% in treated with an OR 0.4 (95%CI 0.12 to 1.36). Can the authors calculate/estimate the sample size they would need to see an effect with this event rate?”

Response: Many thanks for this suggestion. On our power calculations, we estimate that, assuming equal matching in the groups and power set at 80%, the required number of subjects to detect a significant difference in outcome based on the event rates we observed would be 1990. We have added this to the manuscript discussion. 

4) “Table 2 in vaccination status, would be helpful to add categories for more than x2 and more than x3 vaccinations with percentages”

We have edited Table 2 to include the percentages of patients who had received 1, 2, 3, or 4 doses of a Covid-19 vaccination, and separate section for those that received two doses or less / 3 doses or more of a Covid-19 vaccination.

Many thanks for your continued consideration of our manuscript for publication.

Dr Amanda Goodwin

Dr Jonathan Thompson

Professor Ian Hall

---

## [Editor Report · Decision Letter 2]

5 Feb 2023

Evaluation of outpatient treatment for non-hospitalised patients with COVID-19: The experience of a regional centre in the UK

PONE-D-22-19820R2

Dear Dr. Amanda T Goodwin,

We’re pleased to inform you that your manuscript has been judged scientifically suitable for publication and will be formally accepted for publication once it meets all outstanding technical requirements.

Kind regards,

Seung-Hwa Lee

Academic Editor

PLOS ONE
---

## [Editor Report · Acceptance letter]

14 Feb 2023

PONE-D-22-19820R2 

Evaluation of outpatient treatment for non-hospitalised patients with COVID-19: The experience of a regional centre in the UK 

Dear Dr. Goodwin:

I'm pleased to inform you that your manuscript has been deemed suitable for publication in PLOS ONE. Congratulations! Your manuscript is now with our production department. 

Kind regards, 

on behalf of

Dr. Seung-Hwa Lee 

Academic Editor

PLOS ONE